# Mitogen-Activated Protein Kinases (MAPKs) and Cholangiocarcinoma: The Missing Link

**DOI:** 10.3390/cells8101172

**Published:** 2019-09-28

**Authors:** Chaobo Chen, Leonard J. Nelson, Matías A. Ávila, Francisco Javier Cubero

**Affiliations:** 1Department of Immunology, Ophthalmology & ENT, Complutense University School of Medicine, 28040 Madrid, Spain; bobo19820106@gmail.com; 212 de Octubre Health Research Institute (imas12), 28040 Madrid, Spain; 3Department of General Surgery, Wuxi Xishan People’s Hospital, Wuxi 214000, China; 4Institute for Bioengineering (IBioE), School of Engineering, Faraday Building, The University of Edinburgh, Edinburgh EH9 3 JL, Scotland, UK; L.Nelson@ed.ac.uk; 5Hepatology Program, Center for Applied Medical Research (CIMA), University of Navarra, 31008 Pamplona, Spain; maavila@unav.es; 6Centro de Investigacion Biomedica en Red, Enfermedades Hepáticas y Digestivas (CIBERehd), 28029 Madrid, Spain

**Keywords:** cholangiocarcinoma (CCA), mitogen-activated protein kinases (MAPK), cholangiocytes, hepatocytes, epithelial to mesenchymal transition (EMT), cancer-associated fibroblasts (CAFs)

## Abstract

In recent years, the incidence of both liver and biliary tract cancer has increased. Hepatocellular carcinoma (HCC) and cholangiocarcinoma (CCA) are the two most common types of hepatic malignancies. Whereas HCC is the fifth most common malignant tumor in Western countries, the prevalence of CCA has taken an alarming increase from 0.3 to 2.1 cases per 100,000 people. The lack of specific biomarkers makes diagnosis very difficult in the early stages of this fatal cancer. Thus, the prognosis of CCA is dismal and surgery is the only effective treatment, whilst recurrence after resection is common. Even though chemotherapy and radiotherapy may prolong survival in patients with CCA, the 5-year survival rate is still very low—a significant global problem in clinical diagnosis and therapy. The mitogen-activated protein kinase (MAPK) pathway plays an important role in signal transduction by converting extracellular stimuli into a wide range of cellular responses including inflammatory response, stress response, differentiation, survival, and tumorigenesis. Dysregulation of the MAPK cascade involves key signaling components and phosphorylation events that play an important role in tumorigenesis. In this review, we discuss the pathophysiological role of MAPK, current therapeutic options, and the current situation of MAPK-targeted therapies in CCA.

## 1. Introduction

### Cholangiocarcinoma (CCA)

Cholangiocarcinoma (CCA) is a malignant adenocarcinoma characterized by biliary tract differentiation and is the second most common primary liver tumor. Although CCA in most countries is considered a rare form of cancer with an incidence rate of <6 cases per 100,000 of the population [1,2,3], the global incidence of CCA (1/100,000) is steadily on the rise, reaching up to 2.1/100,000 in Western countries, and having increased almost 165% in the United States over the last 30 years period [4,5,6,7]. Risk factors for CCA include chronic viral hepatitis (types B and C), liver cirrhosis, biliary stone disease, chronic infections (infection with *Clonorchis sinensis*, *Opisthorchis viverrini*, and *Opisthorchis felineus* species in Southeast Asia), metabolic syndrome, obesity, and toxic substances (e.g., tobacco and alcohol). In addition, other factors such as parasitic infections, hepatolithiasis, Caroli’s disease, and primary sclerosing cholangitis (PSC) are identified risk factors [8]. Although hepatolithiasis is more commonly found in Asia, it has also been associated with CCA in Western populations [9,10]. In addition, CCA development has also been linked to genetic variants such as DNA repair, inflammation, and carcinogen metabolism [3].

Depending on the anatomical location, CCA can be basically divided into three categories: intrahepatic (iCCA), perihilar (pCCA), and distal (dCCA) [5,11]. The above-mentioned elevation in global CCA incidence is mainly due to the increase in iCCA cases [12]. It is noteworthy that although these three subtypes share common features, they also have important inter- and intra-tumoral differences that may affect the pathogenesis and outcome of the disease [13,14,15,16]. According to a recent study, by using integrated molecular analysis research methods, CCA can be divided into two categories: inflammatory or proliferative. The former is characterized by the activation of inflammatory signaling pathways, overexpression of cytokines, and the activation of the signal transducer and activator of the transcription 3 (STAT3) pathway, which is characterized by the activation of oncogenic signaling pathways including RAS, mitogen-activated protein kinases (MAPKs), and epithelial to mesenchymal transition (EMT) [17]. Consequently, a number of important signaling pathways involved in CCA have been discovered including RAF/mitogen-activated protein kinase (MEK)/MAPK, transforming growth factor-β (TGF-β)/SMADs, phosphatidylinositol 3-kinase (PI3K)/AKT, WNT, interleukin-6 (IL-6)/STAT-3, and NOTCH [18,19]. The MAPKs not only play an important role in the initiation and formation of CCA, but are also closely linked to many CCA-related signaling pathways, as above-mentioned.

## 2. The MAPK Signaling Pathways: ERK1/2, JNK-1/2/3, and p38

There are at least three different MAPK signal transduction pathways that modulate and transduce extracellular signals into the nucleus to induce response genes in mammalian cells including ERK1/2, c-Jun NH2-terminal kinase(JNK)1/2/3, and p38 [20,21]. The ERK kinase family is composed of ERK1 (p44) and ERK2 (p42). The JNK kinase family is composed of JNK1, JNK2, and JNK3. Finally, the p38 MAPKs family can be further divided into two subgroups, p38α and p38β, and p38γ and p38δ, respectively [22,23]. In addition, there are other atypical MAPKs that have unique regulation and function including ERK3/4, ERK7/8, and Nemo-like kinase (NLK) [20,21,24,25].

Growth factor-initiated signaling is associated with the ERK pathway, whereas the JNK and p38 pathways are activated by cytokines, growth factors, environmental stress, and other stimuli (Figure 1). In the p38 pathway, MKK3/6 acts as mitogen-activated protein kinase kinases (MAP2Ks) and is activated by mitogen-activated protein kinase kinase kinases (MAP3Ks) such as apoptosis signal-regulating kinase 1 (ASK1), transforming growth factor-β-activated kinase 1 (TAK1), mitogen-activated protein kinase kinase kinase 1 (MEKK1), and mitogen-activated protein kinase kinase kinase 11 (MLK3). These MAP3Ks also play a role in the JNK pathway, where they target MAP2Ks such as MAP kinase 4 (MKK4)/MAP kinase 7 (MKK7), which activate JNK1/2. In the ERK signaling pathway, ERK1/2 is activated by MEK1/2 and activated by RAF, which is activated by RAS, and RAS is related to the plasma membrane by activation of receptor tyrosine kinases (RTKs), epidermal growth factor receptor (EGFR), vascular endothelial growth factor receptor (VEGFR), and platelet-derived growth factor receptor (PDGFR) [20,26].

### 2.1. JNK/MAPKs

In normal liver, the JNK family is minimally or transiently activated, the latter usually physiological, whereas sustained activation is pathological. In addition to their central role in hepatic physiological responses (such as cell death, apoptosis, and cell cycle regulation) [27,28], JNK plays a carcinogenic role by promoting inflammation, proliferation, invasion, and angiogenesis, depending on the specific context and duration of the activation of the JNK signaling pathway [29,30]. Earlier studies have reported that the JNK signaling pathway plays a significant role in the development of HCC [31,32]. In particular, *Jnk1* seems to be more important in malignant transformation and hepatocellular carcinoma (HCC) development [33,34]. However, during the progression of liver disease, the deletion of *Jnk1* significantly exacerbated apoptosis, compensatory proliferation, and carcinogenesis in experimental chronic liver disease. In turn, *Jnk2* was reported to modulate necrosis and inflammation [35]. Using *Puma* KO mice in diethylnitrosamine (DEN)-induced HCC, HCC formation was suppressed, a phenomenon associated with reduced cell death and compensatory proliferation of hepatocytes. However, such effects were inhibited by the administration of the JNK inhibitor SP600125 [36]. In human HCC cell lines, *Jnk1* activation is associated with tumor size. By knocking down *Jnk1* but not *Jnk2*, proliferation of human HCC occurs via upregulation of *cmyc* and downregulation of *p21* [37].

The JNK signaling pathway has been recently implicated in CCA development. JNK plays an important role in regulating the interaction between different pro-apoptotic and anti-apoptotic proteins in response to both external and internal apoptotic stimuli [38]. In human CCA, a high expression of *TNFA* in cells near CCA lesions was found and phosphorylation of JNK in cholangiocytes (80% in CCA patients) as well as the accumulation of reactive oxygen species (ROS) around the peripheral hepatocytes occurred. Accumulation of ROS leads to caspase-3-dependent apoptosis via JNK. At the same time, Yuan et al. [39], showed that mitochondrial dysfunction and oxidative stress promoted cholangiocellular overgrowth and tumorigenesis. In this study, cholangiocellular proliferation and oncogenic transformation were promoted through the JNK signaling pathway, which was activated by Kupffer cells (KC), subsequently leading to liver damage, ROS, and the paracrine release of TNF. Therefore, KC-derived TNF might promote cholangiocellular cell proliferation, differentiation, and carcinogenesis, suggesting that the ROS/TNF/JNK axis plays a significant role in the development of CCA. Another study that tested the antioxidant activity of using guggulsterone, a steroid found in the resin of the guggul plant (*Commiphora wightii)* in both HuCC-T1 and RBE CCA cell lines, showed that guggulsterone could induce the apoptosis of human CCA cells via ROS-mediated activation of the JNK signaling pathway [40]. Feng et al. [41] reported that JNK exerted its carcinogenic effect in human CCA cells, partially because the mammalian target of the rapamycin (mTOR) pathway was regulated by the induction of glucose-regulated protein 78 (GRP78). They showed that eukaryotic initiation factor-α (eIF2a)/activated transcription factor 4 (ATF4) signaling promoted the accumulation of GRP78; whilst JNK was both promoted by the activation of mTOR and characterized by a high expression of GRP78. These results indicated that GRP78 contributed to the pro-tumorigenic function of JNK in human CCA cells, partly through promoting the eIF2α/ATF4/GRP78 pathway, and through JNK/mTOR signaling. As above-mentioned, the JNK/MAPKs signaling pathway may be a key target in the treatment of CCA; however, more studies are needed to test this possibility.

### 2.2. p38/MAPKs

The p38 kinase plays a key role by promoting proliferation, invasion, inflammation, and angiogenesis in the occurrence and development of CCA [42,43,44,45], also affecting the growth of malignant human CCA cells, and maintaining the phenotype of transformed cells [45,46]. p38δ can be used for differential diagnosis of CCA, since it is not expressed in HCC cells [47]. IL-6 receptors and tyrosine kinases receptors such as Met (c-MET) and the EGFR family members ERBB2 and ERBB1 (EGFR) are key signaling pathways in CCA. Members of the EGFR family, particularly EGFR and ERBB2 (HER-2/neu), are involved in the pathogenesis of CCA [15,48,49]. Dai et al. [50] reported that both p38 and C-MET, the tyrosine kinase receptor for hepatocyte growth factor (HGF), promote the proliferation and invasion of human CCA cells, and p38 promotes CCA formation via sustained activation of C-MET. In addition, using ATO (arsenic trioxide) alone or in combination with metformin to treat CCA cell lines, Ling et al. [42] found that inactivation of p38 by the inhibitor SB203580 or specific siRNA could enhance the anticancer efficacy of single drug or combination of metformin and ATO, especially using ATO alone. Furthermore, the expression of P38δ was upregulated in CCA when compared to normal biliary tissue. Inhibition of P38δ expression by siRNA transfection significantly reduced CCA cell proliferation and invasion. Conversely, overexpression of P38δ in CCA cells can induce an increase in tumor invasiveness.

### 2.3. ERK/MAPKs

Previous studies have reported an upregulation of the RAS-ERK1/2 signaling pathway in CCA [51]. TGF-β transmits signals through SMAD2/3 phosphorylation, but can also transduce signals via RAS-ERK1/2, PI3K, p38, and Rho pathways [52,53,54]. Rac1, a member of the Rho family of small GTPases, plays a pivotal role in the development of CCA [55,56]. Li et al. [57] demonstrated that integrin β6 promoted tumor cell malignant behavior by activating Rac1. In biliary dysplasia, hyperplasia and CCA, TGF-β is overexpressed, a mechanism associated with CCA initiation and formation [51,58,59]. Moreover, ERK1/2 activation has also been linked with TGF-β-induced epithelial to mesenchymal transition (EMT) and cell invasiveness. One study using CCA cell lines indicated that TGF-β enhanced cell invasiveness and mesenchymal features in CCA cell lines [58]. Recently, a study using CCA cell lines (KKU-M213 and HuCCA-1) showed that TGF-β activates ERK1/2 via the SMAD2/3 pathway, enhancing TGF-β activity to promote tumor growth [60]. As above-mentioned, TGF-β can also transduce signals via the RAS-ERK/PI3K pathways [52,53,54]. The PI3K/AKT/PTEN signaling pathway is significantly overexpressed in human CCA tissue [61], therefore it is another important pathway involved in the development of CCA associated with RAS/ERK/MAPKs. PI3K is one of the most important factors in RAS activation and can be involved in the regulation of various functions including cell growth, cell cycle entry, cell survival, cytoskeleton reorganization, and apoptosis [15,62,63] (Figure 1). Loss of phosphatase function of PTEN will result in the constitutive activation of (PI3K/protein kinase B) AKT signaling pathway in CCA. Inhibition of PI3K/AKT signaling effectively blocked the proliferation and invasive behavior of CCA [61]. At the same time, Hyunho et al. [64] using CCA cell lines (SCK and Choi-CK) with inactivation of AKT, showed decreased expression of *BCL2*, and enhanced expression of *BAX*, thereby inducing the apoptosis of resistant cells; whilst the inhibition of ERK1/2 activation did not induce apoptosis, but decreased tumor cell growth. These results indicated that the AKT/ERK1/2 signaling transduction pathway might mediate apoptosis in CCA cell lines. Furthermore, in CCA, the expression of the HGF receptor (HGFR) encoded by the *MET* gene, also known as *cMET*, is increased (12–58%). Activation of *MET* promotes cell invasion and triggers metastasis by directly participating in tumor angiogenesis [65]. Experiments with CCA cell lines indicate that HGFR-dependent CCA cell invasiveness occurs via the AKT/ERK signaling pathway [66,67,68]. In summary, abundant evidence suggests that the ERK signaling pathway is intimately related to cholangiocarcinogenesis. In the future, the ERK signaling pathway may be an effective target for the diagnosis and treatment of CCA; however, more research is needed to confirm this.

### 2.4. ‘Other’ Kinases

Kinases such as transforming growth factor-β-activated kinase 1 (TAK1) belong to the MAP3Ks family. TAK1 is activated in response to cytokines such as TNF, LPS, and TGF-β in multiple cellular systems [69]. Activation of the TGF-β receptor triggers downstream signaling mediated by SMAD family of proteins, which promotes EMT progression in CCA (discussed below). However, phosphorylated TAK1 activates IKK (IκB kinase) and MKK4/7, causing the activation of NF-κB and JNK. Deletion of TAK1 in liver parenchymal cells (both hepatocytes and cholangiocytes) promotes hepatocyte death, inflammation, fibrosis, and hepatocarcinogenesis, coinciding with biliary ductopenia and cholestasis [70,71,72]. ASK1 is another member of the MAPK family, known as MAP3K5 (mitogen-activated protein kinase kinase kinase 5), which can activate the P38 and JNK pathways in response to both oxidative and ER (endoplasmic reticulum) stress as well as stress-induced via inflammatory cytokines (such as TNFα) [73,74,75].

## 3. EMT Transition

Previously, it was generally believed that HCC originated from hepatocytes and that CCA originates from bile duct cells [62]. Moreover, biliary epithelial cells (BECs) may be involved in liver fibrosis [76]. During acute and chronic cholestasis, liver fibrosis involving an increasing number of ductules (the finest ramifications of the biliary tree), is accompanied by polymorphonuclear leukocyte infiltration and increased extracellular matrix deposition, leading to periportal fibrosis and finally PBC (primary biliary cholangitis) [77]. Diaz et al. [78] presented histological data suggesting that EMT occurs in human liver fibrosis, particularly in diseases associated with prominent bile ductular proliferation including biliary atresia and PBC. Other studies indicated that proliferating cholangiocytes play a key role in the induction of fibrosis via EMT [76,79,80]. In CCA, developmental changes seen in cholangiocytes may be due to the activation of hepatic stem/progenitor cells, the proliferation of bile duct cells, or through ductal metaplasia of mature hepatocytes [81,82]. However, alongside the progress seen in histopathological studies, there is now a better understanding of the origins of BECs. Both hepatic progenitor cells (HPCs) [83] and CCA stem cells [62] seem to be capable of differentiating into BECs (Figure 2), whilst CCA stem cells can also originate from HPCs [62,84,85]. Micrographs of in vivo fate-tracking experiments showed the contribution of hepatocytes and HPCs to bile duct repair or hepatocyte injury. In order to cope with biliary tract injury, signals derived from myofibroblasts mediate the activation and proliferation of the HPCs niche. This process can be triggered by NOTCH signal transduction, via JAG1, a NOTCH ligand provided by the stroma of the portal vein, which induces the differentiation of hepatocytes into BECs [86,87,88]. These results indicate that CCA can be derived from common HPCs, which can differentiate into both hepatocytes and cholangiocytes [89,90].

In addition, chronic inflammatory processes can induce the production of a variety of cytokines including TNF, IL-6, TGF-β, and PDGF, promoting the development of CCA by affecting BEC function and proliferation [18,19]. In other words, EMT may also play an important role in CCA progression.

In 1968, Hay [91] noted that during chick embryo development, epithelial cells underwent through several differentiation and dedifferentiation stages and migrated a considerable distance in the body. All of these processes require an interconversion between the epithelial and mesenchymal cell phenotypes (including ‘multipotent’, mesenchymal stem cells) called EMT, and the opposite process, mesenchymal–epithelial transformation (MET). EMT is a special physiological process in which epithelial cells lose polarity and exhibit a physiological process of mesenchymal phenotype. Subsequently, as research progresses, multiple in vivo studies have also been reported. EMT is critical for embryonic development, wound healing, stem cell biology, and plays a key role in the tumorigenic process [92,93]. The first change occurs when epithelial cells lose cell–cell adhesions (at specialized adherent junctions) with the disruption of two major components E-cadherin and β-catenin. Tight junction fractures also occur and result in the loss of apical-basal polarity including cell-extracellular matrix (ECM) contacts. EMT is now known to be mediated by the so-called EMT-activated transcription factors (EMT-ATFs) including SNAIL, TWIST, and ZEB families [92]. EMT-ATFs are key mediators of cell plasticity, playing an important role in all stages of cancer progression from initiation, primary tumor growth, invasion, dissemination, and metastasis to colonization including resistance to therapy [92,94]. The process from primary tumor to metastasis is very complex, and cancer cells need to adapt to multiple and changing environments, which are usually harsh. This plasticity of tumor cells is reflected by the so called back-and-forth transition from differentiation to undifferentiation or partial EMT-associated cancer cell phenotypes, which promote the formation of tumor cells and facilitate their metastasis [95]. In line with CCA development, at the biochemical level, various morphogenetic and environmental signals including TGF-β, WNT, EGFR, and PDGF, inflammatory cytokines and integrin receptor ligands promote EMT [96,97] (Figure 3). Furthermore, TGF-β is also involved in several steps of cancer from tumorigenesis to metastasis, which promotes cancer cell invasion and metastasis through an EMT process, although at later stages [98,99,100,101,102].

The canonical TGF-β pathway relies on the downstream SMAD family [103], which are subsequently phosphorylated after receiving signals from membrane receptors (e.g., EGFR, TNFR, and ILR). The two phosphorylated R-SMADs are then brought together, recruiting SMAD (co-SMAD) and SMAD4 to form a trimer that can be transported from the cytosol to the nucleus [104]. In the nucleus, the heterotrimeric complex binds to DNA in a sequence-specific manner alone or in combination with other coactivators to trigger the transcription of target genes such as SNAIL1, SNAIL2, and other oncogenes [105,106]. The formation of protein complexes between phosphorylated R-SMADs and SMAD4 is a central event in the TGF-β signaling pathway including apoptosis, cell growth and differentiation, and extracellular matrix production [107,108]. Activation of the TGF-β-induced SMADs independent pathway (i.e., the non-canonical pathway) is achieved via TAK1 and JNK/P38 signaling. Interestingly, JNK induced phosphorylation of R-SMAD can activate the EMT programming process. Both JNK and p38 can synergize with SMADs to promote non-canonical TGF-β-induced apoptosis [96,109,110,111]. In SMAD-independent pathways TGF-β can also induce AKT phosphorylation and rapidly activate PI3K, which is possibly involved in SMAD2-induced EMT [112]. Studies on keratinocytes indicate that the PI3K/AKT pathway contributes to completion of TGF-β-induced SMADs-dependent EMT. Similarly, the TGF-β-induced ERK/MAPKs pathway contributes to EMT induction, as ERK is required to remove cell adhesion junctions, which leads to increased cell migration [96,110,113] (Figure 3).

## 4. Tumor-Associated Macrophages (TAMs) and Cancer Associated Fibroblasts (CAFs)

Macrophages and fibroblasts are the main components of infiltrating stromal cells, called tumor-associated macrophages (TAMs) and cancer-associated fibroblasts (CAFs), respectively. CAFs are a heterogeneous cell population; every tumor type consists of tumor- and tissue-specific CAF subpopulations [114,115,116]. Different markers have been used to identify these subgroups including α-smooth muscle actin (α-SMA), fibroblast-specific protein 1 (FSP1), fibroblast activation protein (FAP), and neuron glial antigen-2 (NG2) [116,117]. In human CCA, hepatic stellate cells (HSCs) are thought to be the dominant cellular source of CAFs [118], whilst other studies suggest that CCA-associated CAFs might be derived from portal fibroblasts (PFs) and from bone marrow derived fibrocytes (BMDFs) [119,120,121]. Activation of CAFs is associated with an invasive phenotype of the tumor in CCA [122]. TAMs and CAFs have also been reported to be involved in tumor progression [56,123,124] as well as in the RAS, MAPK, and MET signaling cascades. On the other hand, TAMs can promote the secretion of tumor growth-promoting factors such as TNF-α, IL6, TGF-β, PDGFR, and VEGFR, among others. These factors can usually act on EMT, promoting tumor growth, and metastasis in CCA [56,125,126,127]. Cadamuro et al. [56] suggest that CCA cells recruit CAFs by secreting platelet-derived growth factor D (PDGF-D), promoting CCA cell migration through PDGFR-β, RHO-GTPase, and JNK activation. The recruitment of macrophages is often supported by CAFs and epithelial tumor cells, and is also stimulated by regulatory signaling pathways (e.g., NOTCH and IL6/STAT3) [128,129,130]. In addition, neutrophil migration can be induced by the P13K/AKT and MAPKs signaling pathways, which are manifested by the recruitment of neutrophils in CCA [131]. According to recent studies, CAFs are able to promote CCA progression through interacting autocrine and paracrine signaling pathways, especially in promoting EMT [97,132].

As above-mentioned, the NOTCH signaling pathway (including the NOTCH ligand JAG1) seems to be another particularly important reason for CCA formation. In a murine model of liver cancer driven by transfection of activated forms of *AKT* and *RAS* oncogenes, inhibiting NOTCH and its ligand JAG1 markedly impairs CCA development. According to cell-fate tracing experiments, combined activation of NOTCH and AKT could induce mature hepatocytes to develop into CCA in mice [133,134] (Figure 2). Moreover, a recent study from Thongsom’s group [135] demonstrated that piperlongumine (PL), a naturally-occurring small molecule, could induce CCA cellular apoptosis through ROS-mediated JNK/ERK activation and proteasome inhibition, while the susceptibility of CCA to PL therapy may depend on the activation of AKT. In fact, overwhelming evidence indicates that it is possible to convert hepatocytes into CCA cells when exposed to specific stimuli, and that the key driving factor in this process is NOTCH signal transduction [133,134]. Indeed, activation of the NOTCH signaling pathway induces EMT and increases CCA cell line migration [136,137,138].

Several reports have linked the p38 signaling pathway to the regulation of EMT [139,140,141,142]. As noted previously, there is crosstalk between the p38 and the JNK signaling pathways. The JNK and p38 MAPKs pathways share several upstream regulatory factors, so multiple stimuli may activate both pathways. As previously shown, p38 responds to ROS accumulation to prevent tumorigenesis by promoting growth stagnation and apoptosis [143,144]. ROS can promote TNF-α to activate the JNK signaling pathway and lead to apoptosis. However, TNF-α can also activate NF-κB to inhibit ROS through related genes such as MnSOD (manganese-superoxide dismutase) and FHC (ferritin heavy chain), thereby preventing JNK activation and inhibiting apoptosis [145,146]. While activation of p38 promotes MSK1/2 (mitogen- and stress-activated protein kinase 1/2), and subsequently, NF-κB [147,148]. Furthermore, the JNK signaling pathway has been shown to regulate tricellulin-mediated tight junctions in human pancreatic ductal epithelial cells, indicating that it may play an important role in the differentiation of ductular cells [149,150].

As demonstrated in previous studies, TNF-α may alter the differentiation of hepatocytes in liver disease [151]. In TNF-α-stimulated fibroblast death, JNK activation plays a dual role, inhibiting not only TNF-α-stimulated apoptosis, but also TNF-α-induced necrosis [152,153]. Earlier studies by Nishikawa’s group [151] demonstrated that, within the collagen-rich matrix, TNF specifically stimulated branching morphogenesis associated with the expression of bile duct/ductular markers including cytokeratin 19 (CK-19). Meanwhile, the phosphorylation of JNK and c-Jun increased during collagen gel culture, and TNF-α strongly enhanced bile duct transdifferentiation of hepatocytes, principally by inhibiting hepatocyte differentiation and enhancing ductular morphogenesis. Interestingly, in human liver-fluke-associated CCA, overexpression of TNF-α and NF-κB are characteristic [154].

Several studies have shown that CAFs from human CCA are capable of synthesizing HB-EGF (heparin binding EGF like growth factor) with the activation of EGFR in CCA cells, promoting disruption of cell–cell junctions and increased invasiveness [97,155,156]. EGFR expression and signaling are closely related to the development and progression of CCA [17,157]. Abnormal phosphorylation of EGFR family receptors activates p38, increases COX-2, inhibits apoptosis, and promotes tumor growth [158,159]. Simultaneously, epigenetic regulation of IL-6 promotes the progression of CCA by affecting promoter methylation and gene expression in growth regulatory pathways including EGFR [160]. Another study using sulfated galactans (SG) isolated from the red seaweed *Gracilaria fisheri* inhibited CCA cell proliferation by inhibiting EGFR and ERK phosphorylation, and blocking EGF-induced cell proliferation. These results suggest that SG may interact with EGFR and mediate cell proliferation inhibition, partly by blocking the endogenous activation of the EGFR/ERK/MAPKs signaling pathway [161].

## 5. Biomarkers and Diagnosis

The development of CCA involves genetic alterations of related oncogenes in humans [12,130,162,163,164,165,166,167] (Table 1). Mutations common to tumors all along the chromosome include tumor suppressor genes (*TP53,* and *PTEN*), chromatin-remodeling genes (*ARID1A, ARID1B, BAP1, PBRM1*), and gain of function of oncogenes (*KRAS, BRAF*, and *PIK3CA*). However, mutations in the ERK/MAPKs pathway components are relatively common in CCA. In particular, *KRAS* mutations are associated with a decrease in both progression-free, and overall survival in CCA patients [168]. *KRAS* activating mutations are frequently detected (22%, range 5–57%) [17,169,170], especially in codon 12 hotspots, and have recently been identified as an independent predictor of poor survival after surgery [170,171].

Oncogenes in humans CCA include KMT2C (lysine methyltransferase 2C), roundabout guidance receptor 2 (*ROBO2*), ring finger protein 43 (*RNF43*), paternally expressed 3 (*PEG3*), guanine nucleotide binding protein (G protein) alpha stimulating activity polypeptide 1 (*GNAS*), (BRCA-associated protein 1 (*BAP1*), V-Ki-ras2 Kirsten rat sarcoma viral oncogene homolog (*KRAS*), neuroblastoma RAS viral (*v-ras*) oncogene homolog (*NRAS*), v-raf murine sarcoma viral oncogene homolog B1 (*BRAF*), polybromo 1 (*PBRM1*), AT rich interactive domain 1A (SWI-like) (*ARID1A*), AT-rich interaction domain 1B (*ARID1B*); phosphoinositide-3-kinase, catalytic, alpha polypeptide (*PIK3CA*), phosphatase and tensin homolog deleted on chromosome ten *(PTEN*), protein tyrosine phosphatase non-receptor type 3 (*PTPN3*), cyclin dependent kinase 2a/p16 (*CDKN2A*), mothers against decapentaplegic homolog 4 (Drosophila) (*SMAD4*), tumor protein p53 (*TP53*), isocitrate dehydrogenase 1 (*IDH1*), and isocitrate dehydrogenase 2 (*IDH2*).

As previously mentioned, carcinogenesis in CCA is characterized by recruitment of fibroblasts, ECM remodeling, changes in immune cell migration patterns, and promotion of angiogenesis [172]. Matrix metalloproteinases that degrade and remodel the ECM including MMP1, MMP2, MMP3, and MMP9 are strongly expressed in CCA and are associated with invasive tumors. CCA does not always exhibit a tubular contour, which is easily recognized in histopathological studies; instead, cancer cells often appear nodular or exhibit a form of HCC resembling a neuroendocrine or mixed solid shape [173]. CCA cells usually grow along the wall of the portal vein in connective tissue and directly intrude into the nearby bile ducts.

In practice, immunohistochemical detection of CK-7, CK-19, CK-20, CDX-2, and CA199 biomarkers is often used to identify CCA [12,174]; whilst CK-7 and CAM5.2 biomarkers are strongly expressed in CCA cells. Immunohistochemical detection of human EpCAM on the cell membrane is positive in 90% of cases, while the presence of CK-19 is found in approximately 70% to 80% of cases [164,175,176]. In addition, within the NOTCH signaling pathway, SOX9 plays an important role in the pathophysiology of the biliary tract, and SOX9 has been shown to be associated with the progression of EMT in CCA [177,178]. Unfortunately, none of these immunophenotypes are entirely specific for CCA, since they are often also seen in pancreatic ducts and bile duct primordial cells. Accurate diagnosis of CCA depends mainly on clinical symptoms, histopathology, and imaging findings [173]. Even though recent studies have found that modified branched-chain DNA probes for albumin mRNA (used for in situ detection of albumin expression) have a detection rate of 99% for CCA and 100% for liver cancer [179], these are not detected exclusively in CCA, not even in early stages. As such, most patients are diagnosed with CCA at advanced stages and have limited treatment options, resulting in a poor prognosis [180].

The combination of clinical and biochemical findings including abdominal pain/distension, pruritus, jaundice, and weight loss, imaging techniques like ultrasound (US), computed tomography (CT), magnetic resonance imaging (MRI), ^18^F-fluorodeoxyglucose positron emission tomography (^18^FDG-PET), and serum analysis of non-specific tumor biomarkers such as CA199, CEA (carcinoembryonic antigen), and CD133 [180], usually help to diagnose CCA, but ultimately liver biopsy and pathological diagnosis are used as the gold standard [181].

## 6. Therapy

### 6.1. Surgical Treatment

Currently, surgical resection is still the preferred treatment for CCA. However, only a small number of patients (about 35%) are diagnosed sufficiently early to undergo surgery [182]. In 2012, the 5-year survival rates of surgically treated CCA, iCCA, pCCA, and dCCA were 22–44%, 11–41%, and 27–37%, respectively [4]. However, a recent study including 574 patients with pCCA that underwent right hepatic artery resection and reconstruction, with a perioperative mortality rate of less than 5%, showed a 5-year survival rate of 30% [183]. Contraindications to CCA surgical resection include bilateral, multifocal disease, distant metastases, and comorbidities associated with surgical risk that exceed the patient’s expected surgical benefits. Regional lymph node metastasis is not considered an absolute contraindication to resection, although N1 disease (CCA with regional lymph node metastases including nodes along the cystic duct, common bile duct, hepatic artery, and portal vein) is one of the prognostic factors representing poor prognosis [184,185,186]. Although Bismuth-Corlette type IV pCCA is not considered to be an absolute contraindication to surgical resection, subsequent orthotopic liver transplantation (OLT) is a valid treatment after neoadjuvant radiotherapy [187]. The recurrence rate was 20% and the 5-year survival rate was 68% [187]. However, the selection criteria were strict and 25% to 31% of patients developed disease progression while waiting for OLT and were excluded from the protocol [186,187]. Therefore, single-OLT is not recommended as a CCA monotherapy because of high recurrence rates and long-term survival rates of less than 20% [5,188]. Current guidelines for CCA are still controversial, but due to regional differences between Eastern and Western centers, different surgical resection criteria and surgical strategies are based on applied areas, especially between the Western (US and Europe) and the Eastern centers, with aggressive surgical procedures (including extended hepatectomy and combination with vascular resection in early pCCA). Thus, the actual resection rate of CCA patients has increased, and the prognosis improved in East Asia [183,189,190].

### 6.2. Non-Surgical Treatment: Targeted Therapies

The complexity of the pathogenesis and the apparent heterogeneity of CCA have hindered use of efficacious clinical therapeutics in their management [39,191,192]. The combination of gemcitabine and cisplatin (Table 2) is the first-line chemotherapy for patients with advanced CCA (not suitable for local treatment and surgery) and is also suitable for various anatomical disease subtypes. The median survival of the combination was 11.7 months, and 8.1 months with gemcitabine alone [193,194]. Activation of EGFR leads to downstream activation of MAPKs, however, the efficiency of the EGFR inhibitor erlotinib in clinical trials of CCA is limited according to a Multicenter Phase II clinical trial testing [165,195]. A recent phase III randomized controlled trial showed that by combining erlotinib with gemcitabine/cisplatin, patients had a higher response rate (31% vs. 14%, *p* = 0.004) and longer survival (5.9 months vs. 3.0 months, *p* = 0.049) [196].

Locoregional treatments including radiofrequency ablation, transcatheter arterial chemoembolization (TACE), drug eluting beads-TACE, selective ^90^Y internal artery radiotherapy, and microsphere or external radiation therapy are also applied in CCA [186]. Since technology and equipment have improved, the safety and efficacy of radiotherapy for CCA are better. Advances in these techniques and devices can also increase the radiation therapy dose of biliary tumors and/or improve the protection of non-malignant tissues, thereby improving the therapeutic benefit of radiation therapy for CCA [197,198,199,200].

#### Molecularly Targeted Therapy

Recently, there are several new methods available for CCA including molecularly targeted therapy and immunotherapy. The FGFR inhibitor NVP-BGJ398 exhibits the potential to treat CCA [201,202]. Some clinical studies are currently testing patients with advanced CCA who have FGFR changes (NCT02150967), (NCT01703481), (NCT02699606), and (NCT02265341) [12,203,204]. BGJ398 and another small molecule, FGFR inhibitor PD173072, inhibit the downstream carcinogenic activity of MAPK signaling and FGFR fusion kinase. The clinical efficacy of BGJ398 is currently under assessment in a phase I clinical study for advanced solid tumors, with FGFR1/FGFR2 amplification or FGFR3 mutation (NCT01004224) [165,202,205]. ALK (EML4(echinoderm microtubule-associated protein-like 4)–ALK(anaplastic lymphoma kinase)) and the ROS1 inhibitor Ceritinib are currently being evaluated in a phase II clinical trial of patients with ROS1-positive or ALK-positive advanced-stage pCCA or iCCA (NCT02374489), (NCT02638909), and (NCT02568267) [12]. Anti-MEK therapies in CCA have been tested [206,207,208,209,210]. Binimetinib (MEK162), a potent, selective oral MEK1/2 inhibitor, showed promising evidence of activity in patients with biliary tract cancer [207]. PD901, a MEK inhibitor, has been shown to be effective against CCA harboring KRAS oncogenic mutations via inhibition of cell proliferation and modulation of the tumor microenvironment [208]. Cabozantinib is a multi-kinase inhibitor with anti-MET and anti-VEGFR2 activity. Unfortunately, its activity is limited, median progression-free survival (PFS) is 1.8 months, and has significant toxicity to unselected patients with CCA [209]. Similarly, trametinib, in combination with pazopanib, an oral VEGFR tyrosine kinase inhibitor, in patients with advanced CCA, did not achieve a statistically significant improvement in 4-month PFS over the prespecified null hypothesized 4-month PFS [210]. Sorafenib, a multi-kinase inhibitor with anti-VEGFR and RAF family kinase activity (inhibitor of the RAF/MEK/ERK pathway) [211], combined with the EGFR inhibitor Erlotinib in the treatment of advanced CCA, has been disappointing [212]. Due to chemoresistance, gene therapy capable of selectively inducing human organic cation transporter type 1 (*hOCT1*) in tumor cells was suggested as a potentially useful chemosensitization strategy to improve the response of CCA to sorafenib [213,214]. Again, it has been reported that hot-spot mutations in the *IDH1* and *IDH2* genes are frequent in CCA and can promote epigenetic alterations through the regulation of DNA demethylases activity [215]. Small molecules that inhibit the neomorphic activity of mutant *IDH* can reverse DNA methylation, and, in fact, may promote cancer cell differentiation. Such inhibitors may however also be candidates for the treatment of CCA [216,217,218,219]. Therefore, small molecule inhibitors of mutant IDH1 or IDH2 have demonstrated efficacy, and, in fact, orally available compounds such as AG-120 have entered clinical trials (inhibitor of mutant IDH1) [220]. Preliminary studies treating CCA patients with mutated *IDH1* with AG-120 in Phase I clinical trials (dose escalation and dose amplification cohort studies) showed that this inhibitor was safe and effective (NCT02073994) [221]. Enasidenib, a selective inhibitor of mutant IDH2, is currently being studied in multi-phase I/II trials in patients with *IDH2*-mutant advanced-stage solid tumors including CCA(NCT02273739) [12]. Anetumab ravtansine, an anti-mesothelin antibody-drug conjugate, is undergoing Phase I trials to recruit patients with advanced-stage CCA with abnormal expression of mesothelin [12]. *CDKN2A* is a gene encoding the tumor suppressor proteins p16^INK4A^ and p14^ARF^ and is an important negative regulator of cell cycle progression. Repetitive focal loss of *cdkna2* highlights the potential of CDK4/6 inhibitors in the treatment of CCA including ribociclib and palbociclib [16,166,222]. Several studies have found a subset of CCA patients with high expression of PD-1/PD-L1, suggesting that these patients may be suitable for treatment with PD-1 or PD-L1 inhibitors (NCT02703714, NCT02628067) [223,224,225] (Table 1).

Inhibitors of the JNK signaling pathway may also play an important role in therapy against CCA. To date, a number of small molecule inhibitors have been developed that may regulate specific JNK signaling pathways [31]. One study using the RBE human CCA cell line treated with SP600125, an inhibitor of JNK, demonstrated enhanced TGF-β-induced cell apoptosis in a SMAD4-dependent manner, suggesting that JNK inhibition may be an ideal therapeutic candidate for the treatment of human CCA [233]. However, as JNKs play a pivotal role in normal biological functions, the direct inhibition of JNK itself may also have unexpected side effects. Therefore, the development of highly selective JNK1 or JNK2 inhibitors is essential for the safe treatment of cancer [234]. In this regard, it is important to note that selective inhibition between the individual and their associated JNK proteins may be involved in normal cell functions (such as adaptive immunity and brain development), and thus may lead to adverse cellular effects and unwanted toxicity [235,236,237,238]. Unfortunately, MAPKs-directed therapy is not showing the same clinical efficacy as the experimental models. Thus, further research should also focus on developing substrate specific inhibitors that can make important contributions to mitigate different types of cancer [24] including CCA.

## 7. Conclusions

Recent advances have led to a more profound understanding of the role of the MAPKs signaling pathway in liver neoplastic diseases. Clinical symptoms, histopathology, and gathering of detailed molecular information are of great significance for the diagnosis and treatment of CCA. Utilizing improved experimental models, the important roles of the MAPKs signaling pathway in the pathogenesis of CCA are emerging. Although the exact potential risk factors for CCA have not yet been accurately identified, perhaps the mutation profiles identified are closely related to the underlying causes of CCA. Moreover, some of these mutated proteins can become therapeutic targets for the treatment of CCA. Exciting new research is revealing that liver cancer progression can be attenuated by inhibiting or blocking dysregulated cell transduction components of the MAPKs signaling pathway. If we can elucidate the signal transduction pattern of liver cancer development and carry out specific blockade in certain aspects, it will herald significant breakthroughs and progress in the diagnosis and treatment of CCA.

## Figures and Tables

**Figure 1 cells-08-01172-f001:**
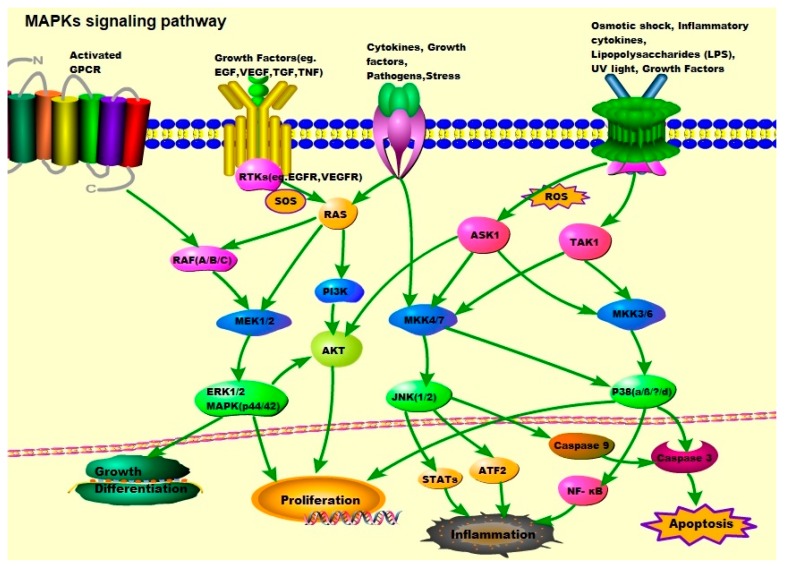
**MAPK signaling pathway: ERK1/2, JNK1/2, and p38**. Activation of ERK begins with the phosphorylation of MEK1/2, followed by activation of tyrosine and threonine residues. Activated RAF binds to and phosphorylates the kinases MEK1/2 as well as activated RAS. The RAS activation occurs at the plasma membrane and is mediated by son of sevenless (SOS), a guanine nucleotide exchange factor (GEF). Signals from cell surface receptors are passed through RAS-GTP to the RAF(A/B/C) and/or PI3K, the latter then activates the PI3K/AKT signaling pathway. RAF also receives signals from the activated ligands’ G protein-coupled receptors (GPCRs). Activated RAF is capable of phosphorylating MEK, and subsequently, the ERK/MAPK signaling pathway. After activation of ERK, ERK1/2 moves to the cytoplasm and nucleus to phosphorylate other proteins. These proteins are responsible for cell regulation, growth, differentiation, and mitosis. JNK is activated in response to cytokines, growth factors, pathogens, stress, etc., and is associated with the transformation of oncogenes and growth factor pathways. Activation of JNK requires dual phosphorylation tyrosine and threonine, the MAP2Ks that catalyze this reaction are known as MKK4 (also known as SEK1) and MKK7. MKK4/7 are phosphorylated and activated by MAP3Ks, TAK1, and ASK1. Tumor necrosis factor (TNF) receptor signaling and ROS might be the major upstream mediators of JNK activation. Abnormal activation of the JNK signaling pathway is linked to the development of cancer, diabetes, inflammatory diseases, and neurodegenerative diseases. The activation of p38 is mediated by upstream kinases, MAP kinase 3 (MKK3), and MAP kinase 6 (MKK6). MKK3/6 are activated by MAP3Ks such as ASK1 and TAK1, which respond to various extracellular stimuli including osmotic shock, inflammatory cytokines, lipopolysaccharides (LPS), UV light, and growth factors.

**Figure 2 cells-08-01172-f002:**
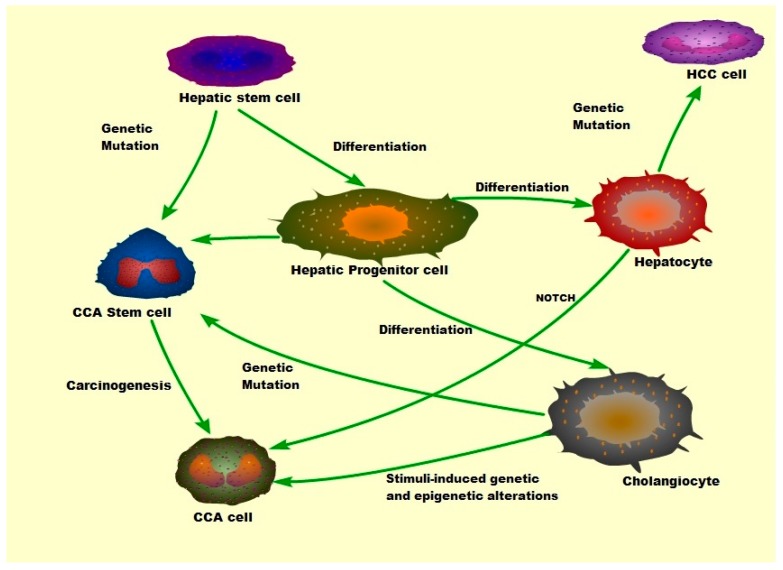
**The hierarchy of cholangiocarcinoma stem cells**. Hepatic stem cells grow and differentiate into hepatic progenitor cells and mature hepatocytes (hepatocytes and cholangiocytes). CCA stem cells are produced by genetic abnormalities of hepatic stem cells, hepatic progenitor cells, or CCA cells, which then grow and differentiate to CCA cells. The NOTCH signaling pathway might induce mature hepatocytes to develop into CCA (in mice). Cholangiocytes can also transform into CCA cells through ROS and inflammation-induced genetic and epigenetic variation.

**Figure 3 cells-08-01172-f003:**
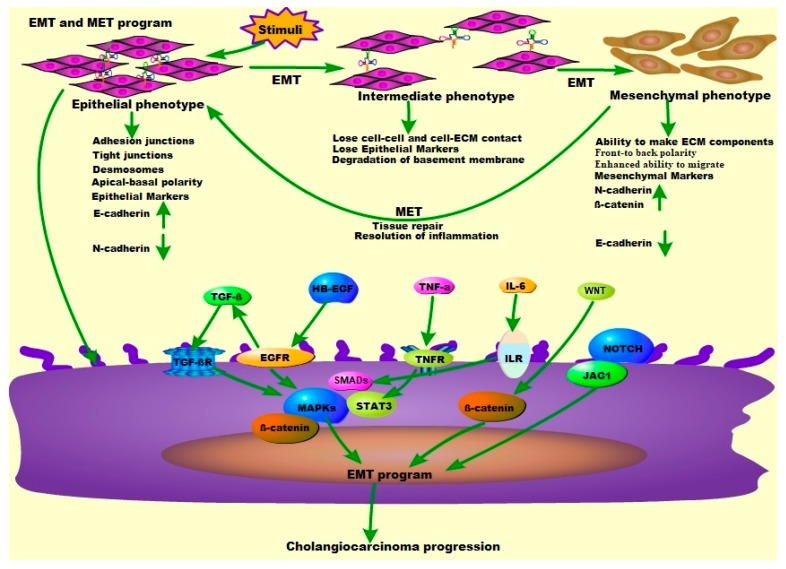
**Schematic diagram of the EMT and MET progress in the CCA program**. Healthy epithelial cells have a tight junction between the apical-basal polarity and interaction with the basement membrane. These cells express markers of epithelial cell lines such as the adhesive connexin E-cadherin. After stimulation, epithelial cells undergo EMT progression (EMT-induced EMT-TFs results in downregulation), lose cell to cell and cell–ECM connections, lose epithelial markers, express interstitial markers (N-CADHERIN, β-CATENIN), obtain front–back polarity, and enhance migration. During tissue repair or inflammation regression, MET is initiated, and the cells reacquire their epithelial phenotype. Cytokines (TGF-β, TNF-α, and IL-6), tyrosine kinase receptors (EGF, HB-EGF), and their receptors (TGFβR, EGFR, TNFR, and ILR) involved in developmental processes play a key role in the induction of the EMT program by activating intracellular signaling pathways (including MAPKs, IL-6/STAT3, NOTCH/JAG1, and WNT), involved in CCA progression and metastasis.

**Table 1 cells-08-01172-t001:** Genetic alterations of related oncogenes in human CCA.

Oncogenes	Tumor Suppressor Genes	Chromatin-Remodeling Genes	Gain of Function of Oncogenes
*MLL3*	*TP53*	*ARID1A*	*KRAS*
*ROBO2*	*PTEN*	*ARID1B*	*BRAF*
*RNF43*		*BAP1*	*PIK3CA*
*PEG3*		*PBRM1*	
*GNAS*			
*NRAS*			
*PTPN3*			
*CDKN2A*			
*SMAD4*			
*IDH1/2*			

**Table 2 cells-08-01172-t002:** Targeted therapies against cholangiocarcinoma.

DRUG	Target	Phase	Identifier
AG-120 [226]	IDH1	I	NCT02073994
Enasidenib [12]	IDH2	I/II	NCT02273739
JNJ-42756493 [227]	FGFR	I	NCT01703481
BGJ398+ PD173072 [165,202,205]	FGFR	I	NCT01004224
NVP-BGJ398 [201,202]	FGFR	II	NCT02150967
Ponatinib [228]	FGFR	II	NCT02265341
Ceritinib [229]	ROS1-ALK	II	NCT02638909
LDK378 (Ceritinib) [229]	ROS2-ALK	II	NCT02374489
Entrectinib [230]	ROS3-ALK	II	NCT02568267
Binimetinib (MEK162) [207]	MEK1/2	I	NCT00959127
Trametinib and Pazopanib [210]	MEK/VEGFR	II	NCT01438554
Cabozantinib [209]	c-MET-VEGFR2	II	NCT01954745
Cisplatin + gemcitabine [193]	EGFR	III	NCT00262769
Gemcitabine/Oxaliplatin + Erlotinib [196]	EGFR	III	NCT01149122
Sorafenib [212]	VEGFR-PDGFR-BRAF	II	NCT00238212
Anetumab ravtansine [12]	anti-mesothelin antibody-drug conjugate	I	NCT03102320
Palbociclib [231]	CDK4/6 Inhibitor	I/II	NCT03065062
Pembrolizumab + GM-CSF [224]	PD1	II	NCT02703714
Pembrolizumab [232]	PD1	II	NCT02628067

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
