# Peer review of "Mitogen-Activated Protein Kinases (MAPKs) and Cholangiocarcinoma: The Missing Link"

_cells, 2019, doi:10.3390/cells8101172_

Round 1
Reviewer 1 Report
I start reading the review and already at the first page it is obvious that authors have two major problems:
- they are mixing up human and mouse gene and protein names. - they make general statements, are unfocused and make strange statements that makes no sense.3.You also end up asking yourself some very basic questions that the authors do not address:
- what is the origin of the cells giving rise to these forms of liver cancer?
- what is the origin of the CAFS in the TME in the different forms of livre cancer discussed?
The authors also do not seem to be very familiar with different type of signaling. A fundamental term in TGF-b signalling is canonical and non-canonical signaling. When discussing mutations, there are certain driver mutations, I also do not find this term.In figure 2 they mention genetic malformation, a term new to me, do the authors mean mutations?
In order to consider publication of this MS , it needs major re-write.
Right now I would strongly argue against publication.
I enclose a copy of MS where I have made some comments, but many more could be made….
I would advise authors to read three recent publications:Rizcvi et al 2018, Nature reviews on clinical Oncol. – overview of driver mutations etc.
Bragazzi et al , Annals of Gastroentereology 2018- origin of cancer cells
Itou et al 2019 human pahtology – origins of CAFs

Author Response
I start reading the review and already at the first page it is obvious that authors have two major problems:
- they are mixing up human and mouse gene and protein names. - they make general statements, are unfocused and make strange statements that makes no sense.
Reply: Thank you for your comments. We apologize for the confusion. We have now revised the manuscript following the rules for gene and protein nomenclatures. The English language has been carefully revised by Dr. Nelson whose mother language is English.
3.You also end up asking yourself some very basic questions that the authors do not address:
- what is the origin of the cells giving rise to these forms of liver cancer?
- what is the origin of the CAFS in the TME in the different forms of livre cancer discussed?
Reply: Thank you. In the revised version of the manuscript we have added information on both aspects.
The authors also do not seem to be very familiar with different type of signaling. A fundamental term in TGF-b signalling is canonical and non-canonical signaling. When discussing mutations, there are certain driver mutations, I also do not find this term.
Reply: The focus of this Review was not on TGFbeta signaling, but we have clarified the canonical and non-canonical TGFbeta signaling pathways in the current version of the manuscript.
In figure 2 they mention genetic malformation, a term new to me, do the authors mean mutations?
Reply: We have also changed ¨malformation¨ for mutation
In order to consider publication of this MS , it needs major re-write.
Right now I would strongly argue against publication.
I enclose a copy of MS where I have made some comments, but many more could be made….
I would advise authors to read three recent publications:
Rizcvi et al 2018, Nature reviews on clinical Oncol. – overview of driver mutations etc.
Bragazzi et al , Annals of Gastroentereology 2018- origin of cancer cells
Itou et al 2019 human pahtology – origins of CAFs
Reply: Thank you for your help in improving the MS. We have taken into consideration your suggestions and changed the manuscript accordingly. The papers mentioned are now included as new references.
Reviewer 2 Report
In their review paper Chen and co-workers discuss the involvement of mitogen-activated protein kinases (MAPKs) in liver cholangiocarcinoma (CCA). Although the authors have excellently summarized and discussed the most relevant literature on this topic, there are some aspects that deserve some additional attention:
The title should be more focalized on the topic of the review specifically mentioning CCA. It would be important to have at the end of the introduction few sentences explaining why the attention has been focalized on the role of MAPKs, taking into account the possible clinical implications. I would suggest of adding at the end of each paragraph a statement summarizing the main implication of each group of MAPKs in CCA. The relationship between the growing knowledge on the implication of MAPKs in the pathogenesis of CCA and the use of specific treatment should be better explained.Author Response
In their review paper Chen and co-workers discuss the involvement of mitogen-activated protein kinases (MAPKs) in liver cholangiocarcinoma (CCA). Although the authors have excellently summarized and discussed the most relevant literature on this topic, there are some aspects that deserve some additional attention:
The title should be more focalized on the topic of the review specifically mentioning CCA. It would be important to have at the end of the introduction few sentences explaining why the attention has been focalized on the role of MAPKs, taking into account the possible clinical implications. I would suggest of adding at the end of each paragraph a statement summarizing the main implication of each group of MAPKs in CCA. The relationship between the growing knowledge on the implication of MAPKs in the pathogenesis of CCA and the use of specific treatment should be better explained.
Reply: Thank you for your comments. We have changed ¨liver cancer¨ for ¨cholangiocarcinoma¨ in the title. We have also added paragraphs to summarize the main aspects of the MAPK pathway in CCA, and we added more data on the new therapeutic avenues of MAPK-directed therapy.
Reviewer 3 Report
Dear Authors,
The authors wrote a review about the potential link of mitogen-activated protein kinases (MAPK) and liver cancer/cholangiocarcinoma. The paper is well written and easy to read. The first part of the review presents general information about cholangiocarcinoma (CCA) and the pathway that could be the cause of CCA. MAPK pathway are presented in detail, mainly 3 major’s pathways: JNK, p38 and ERK. CCA formation is due to an epithelial to mesenchymal transition, that are related with the MAPK activities. Therapies for cholangiocarcinoma can be surgically or non-surgically treated, with drugs that can block growth factors receptors or MAPK. The poor positive results obtained on human studies, compared to the animal model, indicates the difficulty to translate animal results to humans.
I have some questions and comments about the manuscript:
Major:
The title mentioned liver cancer, when the focus is the cholangiocarcinoma. Cholangiocarcinoma should be mentioned in the title. In the part 5, the authors mentioned biomarkers and diagnosis, but very few is related with MAPK. The authors should explain more the relation between the mutation/overactivity of the MAPK and some biomarkers used to detect the CCA (eg https://www.nature.com/articles/labinvest2010161.pdf?origin=publication_detail)
Minor:
The length of the abstract is 229 words, when it should be 200 words maximum. The authors should correct it. In the conclusion, the authors use too much the word therefore in a row, in few sentences. This should be modified.
Author Response
The authors wrote a review about the potential link of mitogen-activated protein kinases (MAPK) and liver cancer/cholangiocarcinoma. The paper is well written and easy to read. The first part of the review presents general information about cholangiocarcinoma (CCA) and the pathway that could be the cause of CCA. MAPK pathway are presented in detail, mainly 3 major’s pathways: JNK, p38 and ERK. CCA formation is due to an epithelial to mesenchymal transition, that are related with the MAPK activities. Therapies for cholangiocarcinoma can be surgically or non-surgically treated, with drugs that can block growth factors receptors or MAPK. The poor positive results obtained on human studies, compared to the animal model, indicates the difficulty to translate animal results to humans.
I have some questions and comments about the manuscript:
Major:
The title mentioned liver cancer, when the focus is the cholangiocarcinoma. Cholangiocarcinoma should be mentioned in the title. In the part 5, the authors mentioned biomarkers and diagnosis, but very few is related with MAPK. The authors should explain more the relation between the mutation/overactivity of the MAPK and some biomarkers used to detect the CCA (eghttps://www.nature.com/articles/labinvest2010161.pdf?origin=publication_detail)
Reply: We have changed ¨liver cancer¨ for ¨cholangiocarcinoma¨ in the title. We have added more data on biomarkers according to the suggestion of the Reviewer. Thank you.
Minor:
The length of the abstract is 229 words, when it should be 200 words maximum. The authors should correct it. In the conclusion, the authors use too much the word therefore in a row, in few sentences. This should be modified.
Reply: Thank you for your comments. We have reduced the length of the abstract and removed ´therefore´ when overused.
Round 2
Reviewer 1 Report
The authors have responded to the comments in a satisfactory manner, and the MS has improved considerably, compared with the previous version. Unfortunately some mistakes that needs correction still exist.
Line 108 - transforming growth factor b lacks a hypen and should read “transforming growth factor-b
Line 242- “,while more researchered is needed to confirm it” should be changed to “,while more research is needed to confirm this”
Line 248 “SMADs family” should read “SMAD family” ( as it stand now it is double plural form)
Line 264 “increased matrix deposition” insert extracellular to read “increased extracellular matrix deposition”
Line 318 “SMADs family” to “SMAD family
Line 352. The first part in the new section on CAFs is not optimal, a few points:
- FSP1 – is more and more realized to also be present on immune cells, not typically a major subpopulation.
- NG2 is a pericyte marker which can be present on pericyte-derived CAFs
- FAP is short for fibroblast activation protein NOT fibroblasts activating protein.
I suggest a rewrite:
CAFs are heterogeneous and every tumor type consists of tumor- and tissue-specific CAF subpopulations. Markers used to identify these subgroups include a-smooth muscle actin (a-SMA), fibroblast-specific protein 1 (FSP1), Fibroblasts activation protein ( FAP) Neuron glial antigen-2 (NG2) etc.
Two reent references the authors shoudl consider citing are:
Cancer-associated fibroblasts: how do they contribute to metastasis?
Kwa MQ, Herum KM, Brakebusch C.
Clin Exp Metastasis. 2019 Apr;36(2):71-86. doi: 10.1007/s10585-019-09959-0. Epub 2019 Mar 7. Review.
Cancer-associated fibroblasts in desmoplastic tumors: emerging role of integrins.
Zeltz C, Primac I, Erusappan P, Alam J, Noel A, Gullberg D.
Semin Cancer Biol. 2019 Aug 12. pii: S1044-579X(19)30038-0. doi: 10.1016/j.semcancer.2019.08.004. [Epub ahead of print] Review.
Author Response
Thank you very much for all your comments. We have corrected (in red in the revised version of the manuscript) all your points, including the changes in the CAFs section which we agree are important. Additionally, we have read and added the suggested references. Altogether, we believe that the manuscript has improved and we are very grateful for your help.
Reviewer 3 Report
Dear Authors,
The authors wrote a review about the potential link of mitogen-activated protein kinases (MAPK) and liver cancer/cholangiocarcinoma. The paper is well written and easy to read. The first part of the review presents general information about cholangiocarcinoma (CCA) and the pathway that could be the cause of CCA. MAPK pathway are presented in detail, mainly 3 major’s pathways: JNK, p38 and ERK. CCA formation is due to an epithelial to mesenchymal transition, that are related with the MAPK activities. Therapies for cholangiocarcinoma can be surgically or non-surgically treated, with drugs that can block growth factors receptors or MAPK. The poor positive results obtained on human studies, compared to the animal model, indicates the difficulty to translate animal results to humans. The authors improved the manuscript by following the reviewer’s suggestions, but there are still some errors to be corrected or explained.
I have some questions and comments about the manuscript:
Major:
The gene nomenclature should be respected: page 6, line 237, c-MET should be replaced by MET. In table 2, the official name of some genes is incorrect: g KM12C for MLL3. Names should be verified and corrected.Minor:
In the table 2, some genes are in blue and others in red. Could the authors explain the reason in the table legend?Author Response
Thank you for your comments and suggestions. We have changed the gene nomenclature for KM12C. However, in the literature you can find both KM12C and MLL3. In addition we have changed in page 6 c-MET for MET.
We apologize for the confusion with the colors in the previously revised version of the manuscript. We have now changed only to ´red´ to indicate changes.